# Laccase-Mediated Oxidation of Phenolic Compounds from Wine Lees Extract towards the Synthesis of Polymers with Potential Applications in Food Packaging

**DOI:** 10.3390/biom14030323

**Published:** 2024-03-08

**Authors:** Panagiotis E. Athanasiou, Christina I. Gkountela, Michaela Patila, Renia Fotiadou, Alexandra V. Chatzikonstantinou, Stamatina N. Vouyiouka, Haralambos Stamatis

**Affiliations:** 1Laboratory of Biotechnology, Department of Biological Applications and Technology, University of Ioannina, 45110 Ioannina, Greece; p.athanasiou@uoi.gr (P.E.A.); mpatila@uoi.gr (M.P.); renia.fotiadou@gmail.com (R.F.); alexandra_xatzi@hotmail.com (A.V.C.); 2Laboratory of Polymer Technology, School of Chemical Engineering, National Technical University of Athens, Zographou Campus, 15772 Athens, Greece; cgkountela@mail.ntua.gr (C.I.G.); mvuyiuka@central.ntua.gr (S.N.V.)

**Keywords:** oxidoreductase, winery by-products, phenolic polymers, polymerization, enzymatic oxidation, chitosan films, antioxidant activity

## Abstract

Laccase from *Trametes versicolor* was applied to produce phenolic polymeric compounds with enhanced properties, using a wine lees extract as the phenolic source. The influence of the incubation time on the progress of the enzymatic oxidation and the yield of the formed polymers was examined. The polymerization process and the properties of the polymeric products were evaluated with a variety of techniques, such as high-pressure liquid chromatography (HPLC) and gel permeation chromatography (GPC), Fourier-transform infrared (FTIR) and nuclear magnetic resonance (NMR) spectroscopies, differential scanning calorimetry (DSC), and thermogravimetric analysis (TGA). The enzymatic polymerization reaction resulted in an 82% reduction in the free phenolic compounds of the extract. The polymeric product recovery (up to 25.7%) and the molecular weight of the polymer depended on the incubation time of the reaction. The produced phenolic polymers exhibited high antioxidant activity, depending on the enzymatic oxidation reaction time, with the phenolic polymer formed after one hour of enzymatic reaction exhibiting the highest antioxidant activity (133.75 and 164.77 μg TE mg^−1^ polymer) towards the ABTS and DPPH free radicals, respectively. The higher thermal stability of the polymeric products compared to the wine lees phenolic extract was confirmed with TGA and DSC analyses. Finally, the formed phenolic polymeric products were incorporated into chitosan films, providing them with increased antioxidant activity without affecting the films’ cohesion.

## 1. Introduction

Laccases (EC 1.10.3.2) are oxidoreductases that can catalyze the oxidation of several compounds, by producing water from molecular oxygen [1]. Many different laccases are produced from various organisms such as bacteria and fungi [2,3,4,5]. One of the most used laccases is the laccase derived from the mushroom *Trametes versicolor* (*T. versicolor*). Laccase from *T. versicolor* (TvL) is well characterized as a biocatalytic tool for the oxidation of different substrates, depending on its redox potential. Moreover, TvL is applied in enzymatic oligomerization/polymerization of phenolic compounds, as an alternative to chemical methods [1,6,7], offering the advantages of the absence of toxic chemicals, controlled reaction conditions, and a wide range of the initial substrates [8]. This oligo- or polymerization reaction is conducted via two main stages: the enzyme-catalyzed generation of phenoxy radicals and the spontaneous non-enzymatic oxidative coupling [9]. In the first step, laccase catalyzes the oxidation of the phenolic compounds, creating phenoxy radicals and quinones. In the second, the non-enzymatic step, the phenoxy radicals or the quinones form C-O or C-C covalent bonds via oxidative phenolic coupling, oxidative coupling, or oxidative condensation. This procedure leads to the production of different homomolecular or heteromolecular oligomers or polymers [9,10,11]. Moreover, in the presence of other non-laccase reactants, laccase can promote the synthesis of heteromolecular dimers or oligomers between a phenolic compound and a non-laccase substrate. This procedure may lead to the formation of phenolic, quinonoid, or quinonimine structures [12]. Some of the indicative laccase-promoted polymerization reactions are summarized in Figure 1. Several studies aim to produce new biopolymers, derived from the enzymatic oxidation of phenolics, to improve the biological or physicochemical properties of their monomer precursors [13,14]. These phenolic polymers could be used as alternatives to their monomer precursors due to their high stability and antioxidant activity. For example, the dimerization of ferulic acid by laccase leads to increased antioxidant activity [13], while the polymerization of flavonoids amplifies their thermal stability [15].

However, most studies report the enzymatic polymerization of individual compounds to form dimers, trimers, or oligomers [7,12,16,17]. Only a small part of the recent literature presents the polymerization of a mixture of different phenolic compounds or the polymerization of phenolics in natural extracts [16,18,19]. Natural extracts constitute an area of great research interest due to their high content of bioactive compounds [20,21]. The antioxidant, antimicrobial, and other biological activities of extracts are often attributed to their high phenolic content [22,23,24]. In this context, Su and colleagues reported the laccase-mediated synthesis of polymers from a bamboo extract [18]. The authors claimed that laccase promoted the bonding of phenolics, leading to the production of hard bamboo tablets. However, the polymers were not further investigated for their biological and structural properties. In another work, propolis and poplar bud extracts were treated with laccase to enhance their antioxidant activity [25]. The obtained products were characterized for their antioxidant activity, concluding that a lower reaction time led to a higher antioxidant activity than the parental extract. Recently, the laccase-catalyzed synthesis of polymers from cork and grape extracts was reported [19]. The formed polymers exhibited remarkable antioxidant and antiaging properties compared to the initial extract. The authors also provided some information on the composition and structure of the oxidation polymeric products. Thus, due to the restricted knowledge of the use of natural extracts as the phenolic source for polymerization reactions, and the further characterization of the resulting polymers, research in this field constitutes an area of great interest in synthesizing novel bio-products.

One potential application of these biopolymers could be their incorporation in food packaging films, reinforcing their parental properties. In recent years, the need to replace plastic films has arisen due to the great environmental problem. Biopolymers such as starch, gelatin, and chitosan (CS) are derived from natural sources and are used as alternatives for food packaging film production due to their characteristics and their high biodegradability [26]. The use of phenolic compounds and natural extracts as additives in food packaging films is well known due to their high antioxidant and antimicrobial activity [26,27,28,29]. Furthermore, only a small part of the literature reports the use of grape or winery by-product extracts as additives in biopolymer films [27,30,31]. However, the stability of the extract-/phenolic-enriched films may be low due to the oxidation of the incorporated phenolic compounds, while small phenolic antioxidants could be released from the polymer matrix during contact with water [32]. This problem could be addressed by replacing natural extracts with phenolic polymers; this may improve the stability of the films by reducing the releasing effect and providing them with sufficient antioxidant activity.

Considering all that mentioned above, the present work reports the laccase-catalyzed polymerization of phenolic compounds of wine lees extract (WLE) towards the production of novel functional polymers. Wine lees, as one of the most abundant and important by-products of the winemaking industry, are an excellent source of phenolic compounds, with high antioxidant and other biological activities [23,33,34,35]. The phenolic polymerization reaction of the WLE was studied for different time intervals to produce phenolic polymers with tunable characteristics. The polymerization reaction and the formed polymeric products were analyzed with spectroscopic and analytical techniques to monitor the polymerization process and to investigate the characteristics of the newly formed products, such as thermal stability. Lastly, the different polymeric products were incorporated in CS films to enrich them with antioxidant activity, indicating that they can be effectively used as alternative additives in biodegradable food packaging films. To our knowledge, this is the first time that high molecular weight phenolic polymers, derived from the enzymatic oxidation–polymerization of a natural extract, are incorporated into CS films.

## 2. Materials and Methods

### 2.1. Chemicals and Reagents

Laccase from *Trametes versicolor* (0.78 U mg^−1^), Folin–Ciocalteu’s phenol reagent, chitosan (75–85% deacetylated, low molecular weight, 20–300 cP), acetic acid (99.8%), 2,2-diphenyl-1-picrylhydrazyl (DPPH), and 2,2′-azino-bis (3-ethylbenzothiazoline-6-sulphonic acid) diammonium salt (ABTS) were purchased from Sigma-Aldrich (St. Louis, MO, USA). Potassium peroxydisulfate was purchased from Merck KGaA (Darmstadt, Germany). Gallic acid hydrate (GA) and 6-hydroxy-2,5,7,8-tetramethylchroman-2-carboxylic acid (Trolox) were obtained from Tokyo Chemical Industry Co., Ltd. (Tokyo, Japan). Methanol (HPLC grade), acetonitrile (HPLC grade), water (HPLC grade), potassium bromide, and glycerol were purchased from Fisher Scientific Co. (Loughborouch, UK). Ethanol (99.8%), acetone (99.5%), and sodium carbonate were purchased from Riedel de Haen (Charlotte, NC, USA). n-hexane and N, N-dimethylformamide (≥99.9%) were purchased from Carlo-Erba (Emmendingen, Germany). Methyl alcohol-D4 was purchased from Deutero (Kastellaun, Germany). Double-distilled water (ddH_2_O) was used for all experiments.

### 2.2. Wine Lees Extraction

Wine lees were recovered from a wine produced in a local winery (region of Epirus, Greece). Syrah, Merlot, and Cabernet red grapes (60, 30, and 10%, respectively), originating from the Nemea wine region (Peloponnese, Greece), were destemmed and pressed, and fermentation was conducted by adding to the must 20 g/hL of *Saccharomyces cerevisiae* Mycoferm-Pro (Ever S.R.L, Pramaggiore VE, Italy). The fermentation was carried out at 18 °C. The wine lees were collected from the bottom of a stainless steel wine stabilization tank, washed twice with distilled water, and centrifuged (9500 rpm, 10 min). The obtained pellet was used for phenolic extraction. The ultrasound-assisted extraction of wine lees was carried out in 70% ethanol under the previously optimized conditions: solid-to-solvent ratio 1:5, 20 min extraction time, and ultrasound power at 200 Watt. The ethanol was evaporated through rotary evaporation (Buchi Rotavapor R-114, Büchi Labortechnik AG, Flawil, Switzerland), and the extract was freeze-dried. The extract was stored at −20 °C until use.

### 2.3. TvL Activity and Stability

The enzymatic activity of TvL was determined using ABTS as the substrate [36]. The reaction mixture consisted of 1 mM ABTS and 50 μg mL^−1^ TvL in 0.1 M phosphate buffer pH 5.7. The oxidation of the ABTS was measured spectrophotometrically at 405 nm (*ε*_405_ = 36,000 M^−1^ cm^−1^) for 5 min. One unit of enzymatic activity (U) was determined to be the amount of enzyme that catalyzed the oxidation of 1 μmol of ABTS per minute.

The residual activity of TvL was measured after incubation at 30 °C to determine its stability during the polymerization reaction. Briefly, a TvL solution of 0.5 mg mL^−1^ in buffer was incubated at 30 °C, and appropriate aliquots were taken at 0, 1, 2, 4, 6, and 24 h to determine the activity, as described above. The results are expressed as % residual activity through the following Equation (1):(1)$$Residual\;activity(\%)=\frac{A_{0}-A_{t}}{A_{0}}*100$$
where *A*_0_ is the initial activity of TvL at *t* = 0 and *A_t_* is the activity of TvL at different time intervals.

### 2.4. Enzymatic Synthesis of Phenolic Polymers

The enzymatic oxidation of the WLE and the further synthesis of the phenolic polymers was carried out according to a protocol developed for standard phenolic compounds, with some modifications [16]. The reaction mixtures (50 mL) contained 2.5 mg mL^−1^ WLE, 0.5 mg mL^−1^ TvL, and phosphate buffer (0.1 M, pH 5.7). The reaction was carried out in round bottom flasks at 30 °C and under continuous stirring at 140 rpm for 1, 2, 6, and 24 h. After that, the reaction mixtures were transferred into 50 mL falcon tubes and left overnight at 4 °C to facilitate the precipitation of the polymeric products. Then, the samples were centrifuged at 9500 rpm for 10 min at 4 °C, and the precipitants were washed at least three times with water: acetone mixture (3:1) until the supernatant was clear. One more washing step with water was also performed to remove the excess acetone. Finally, the polymeric products were freeze-dried, weighed, and stored at −20 °C until used. A blank reaction without enzyme was also prepared to investigate the self-oxidation of the extract phenolics during the process. The produced polymeric products are named P1h, P2h, P6h, and P24h corresponding to 1, 2, 6, and 24 h of reaction, respectively. All experiments were conducted in triplicate.

### 2.5. Dephenolization of the Extract during Polymer Synthesis

The total phenolic content (TPC) of the reactions, before the enzymatic polymerization and after the enzymatic polymerization, was estimated with the Folin–Ciocalteu method, as described by Spyrou et al., and adapted at a total volume of 200 μL [6]. A standard curve of gallic acid was used to express the results as mg of gallic acid equivalents (GAE) per mL (mg GAE mL^−1^). The dephenolization yield was calculated through the following Equation (2):(2)$$Dephenolization\;(\%)=\frac{{TPC}_{i}-{TPC}_{s}}{{TPC}_{i}}*100$$
where *TPC_i_* is the initial total phenolic content of the reaction mixtures and *TPC_s_* is the total phenolic content in the supernatant after the enzymatic polymerization. All experiments were conducted in triplicate.

### 2.6. Analytical and Spectroscopic Characterization

#### 2.6.1. Ultraviolet–Visible Spectroscopy

The alterations in the ultraviolet–visible (UV-Vis) spectra of the extract, before and after different durations of the enzymatic treatment of the extract with TvL, were monitored in the range of 250–800 nm on an Elisa reader (Thermo Scientific, Waltham, MA, USA), on 96-well UV-Elisa plates. The samples were centrifuged (12,000 rpm, 2 min) and diluted with buffer (1:4), and an aliquot of 200 μL was used for spectra recording.

#### 2.6.2. HPLC Analysis

Chromatographic separation of the phenolic compounds of the extract in the reaction buffer before and after the enzymatic treatment was performed using a high-pressure liquid chromatography system (Shimadzu, Tokyo, Japan) equipped with a photodiode array detector and a Kinetex Evo C18 reversed-phase column (5 μm, 250 × 4.6 mm) with a guard column, Gemini-NX C18 (4 × 3.0 mm) (Phenomenex, Torrance, CA, USA). Samples were dissolved in 10% acetonitrile and filtered through a 0.22 µm nylon membrane syringe filter. The column temperature was set at 30 °C, and the flow rate and injection volume were 1 mL min^−1^ and 20 μL, respectively. The mobile phase consisted of acetonitrile (A) and water (B, with 0.1% acetic acid) with gradient elution from 10–90% at 0–5 min to 16–84% B at 5–18 min, 18–82% B at 18–26 min, 28–72% B at 26–31 min, 40–60% B at 31–40 min and 10–90% B at 40–43 min; the total program time was 43 min. Phenolic compounds were identified based on the retention time and absorption profile spectra of the reference substances. All comparisons were carried out at the absorption maxima of the reference compounds, and the concentration of each compound was calculated from standard curves in terms of μg mL^−1^. The results are expressed as % reduction, which is calculated using the following Equation (3):(3)$$Reduction(\%)=\frac{C_{i}-C_{f}}{C_{i}}\times 100$$
where *C_i_* represents the concentration of each compound in the extract before the enzymatic treatment and *C_f_* is the concentration of each compound in the reaction mixture after the enzymatic treatment.

#### 2.6.3. NMR Analysis

^1^H NMR experiments were conducted using a Bruker AVANCE 500 MHz spectrometer at 298 K, equipped with a broadband inverse probe (Bruker Biospin, Rheinstetten, Germany). Samples (8 mg mL^−1^) were dissolved in deuterated methanol (CD_3_OD). Spectral data were processed using TopSpin 4.1.3 software.

#### 2.6.4. Fourier-Transform Infrared Spectroscopy

Fourier-transform infrared (FTIR) was used to investigate the polymerization stages of wine lees extract phenolics. The spectral measurements were performed at the 400–4000 cm^−1^ range, using 64 scans and a 4 cm^−1^ resolution, in a Jasco FT/IR 4700 (Jasco Co., Ltd., Tokyo, Japan) spectrometer. The samples were analyzed after mixing with KBr to form pellets including 1% (*w/w*) of the sample.

#### 2.6.5. Gel Permeation Chromatography Analysis

Gel permeation chromatography (GPC) was carried out using the Agilent 1260 Infinity II instrument (Agilent Technologies, Santa Clara, CA, USA), equipped with a guard column (PLgel 5 μm) and two PLgel MIXED-D 5 μm columns (300 × 7.5 mm). Elution was performed with N, N-dimethylformamide at a 1 mL·min^−1^ flow rate. The analysis was performed using the Agilent 1260 Infinity II refractive index detector (RID) (G7162A). The instrument was calibrated with polystyrene standards of molecular weight from 162 to 500,000 g mol^−1^ (EasiVial PS-M 2 mL, Agilent Technologies, Santa Clara, CA, USA). It should be noted that a universal calibration was not applied, due to a lack of Mark–Houwink (K and a) constants for the specific polymer–solvent system.

#### 2.6.6. Differential Scanning Calorimetry Measurements

Differential scanning calorimetry (DSC) measurements were performed in a Mettler DSC 1 STARe System (Mettler Toledo, Columbus, OH, USA) under N_2_ flow (10 mL·min^−1^). A small quantity of each sample (8–25 mg, depending on the sample morphology) was isothermally heated at 100 °C to remove any residual water or solvents and then heated from 0 to 400 °C at a rate of 10 °C min^−1^.

#### 2.6.7. Thermogravimetric Analysis

Thermogravimetric analysis (TGA) was conducted in a Mettler TGA/DSC 1 thermobalance (Mettler Toledo, Columbus, OH, USA) from 30 to 600 °C at a 10 °C min^−1^ heating rate under N_2_ flow (20 mL min^−1^). Before the measurement, the samples were isothermally heated at 100 °C to remove any residual water or solvents. The onset decomposition temperature was defined as the temperature at 5% weight loss (*T*_d,5%_), the degradation temperature (*T*_d_) was determined at the maximum rate of weight loss, and the char yield was defined as the % residue at 600 °C.

### 2.7. Formation of Polymers-Enriched CS Films

A CS solution was prepared by dissolving 3 g of CS in 200 mL of a 1% (*v/v*) acetic acid solution at 75 °C for 30 min under continuous stirring [37]. Then, glycerol, 3% (*w/w,* concerning chitosan), was added as a plasticizer under stirring for 10 min. Then, 10 mL of the chitosan solution was mixed with the polymeric products. Three different loading concentrations were tested. The consistency of the different produced films for each polymeric product is summarized in Table 1. The solutions were cast in 50 mm plastic Petri dishes and dried at 35 °C for 4 days. CS films without polymer addition were prepared as control samples.

### 2.8. Antioxidant Activity

The antioxidant activity of the polymeric compounds was tested against ABTS and DPPH free radicals.

The ABTS assay was conducted according to Adelakun et al. with some modifications [13]. In brief, ABTS and potassium peroxydisulfate aqueous solutions were mixed to achieve final concentrations of 7.0 and 2.45 mM, respectively. The mixture was incubated in darkness for 16–18 h and then diluted with ddH_2_O to obtain an absorbance of 0.700 ± 0.020 at 734 nm. Then, 30 μL of the methanolic polymer solutions were mixed with 270 μL of the diluted ABTS^•+^ to achieve final concentrations of 10–100 μg mL^−1^. The absorbance of the samples was measured at 734 nm after 30 min of incubation in darkness.

The DPPH assay was conducted according to previous works [6,20]. A methanolic solution of DPPH (0.1 mM) was prepared, and the tested concentration of the polymers was 10–150 μg mL^−1^. The reaction mixtures were incubated in darkness for 30 min, and the absorbance of the samples was monitored at 517 nm.

The results for both assays were expressed as μg Trolox equivalents (TE) per mg of dry polymers, using Trolox standard curves.

The same methods were used to determine the antioxidant activity of the CS-polymers films. The films were cut into 1 × 1 cm (~9 mg) and 0.6 × 0.6 cm (~4 mg) pieces for ABTS and DPPH assays, respectively. For the ABTS assay, 1 mL of the diluted ABTS^•+^ was mixed with the film’s pieces, and the absorbance was monitored at 734 nm after 5, 10, 20, and 30 min of incubation in darkness. For the DPPH assay, 1 mL of a methanolic DPPH solution (0.03 mM) was mixed with the film’s pieces, and the absorbance was monitored at 517 nm after 30 min of incubation in darkness.

The results are presented as % antioxidant activity after 30 min of incubation for both assays using the following Equation (4):(4)$$Antioxidant\;activity(\%)=\frac{A_{control}-A_{sample}}{A_{control}}\times 100$$
where *A_control_* represents the absorbance of the control sample (only the free radical without sample interaction) and *A_sample_* represents the absorbance of the free radical after the interaction with the tested sample. All experiments were conducted in triplicate.

### 2.9. Moisture Content, Water Swelling, and Water Solubility Assays

The moisture content (MC) and water solubility (WS) of the films were determined according to Liu et al. [38], and the water swelling (WSw) degree of the films was evaluated according to Kahya et al. with minor modifications [28]. In our case, the drying time of the films was adapted to 4 h because no significant weight changes were observed after 3 h of drying at 105 °C. Furthermore, the water immersion of each film piece was conducted in 10 mL of ddH_2_O for 24 h.

The MC of the films was calculated using the following Equation (5):(5)$$MC=\frac{W_{1}-W_{2}}{W_{1}}\times 100$$
where *W*_1_ is the initial weight of the film and *W*_2_ is the weight of the dried film.

The *WSw* degree was calculated using the following Equation (6):(6)$$WSw=\frac{W_{1}-W_{2}}{W_{1}}\times 100$$
where *W*_1_ is the weight of the water-immersed film and *W*_2_ is the weight of the dried film.

The *WS* was calculated using the following Equation (7):(7)$$WS=\frac{W_{1}-W_{2}}{W_{1}}\times 100$$
where *W*_1_ is the weight of the dried film before the water immersion and *W*_2_ is the weight of the dried film after water immersion.

All experiments were performed in triplicate.

## 3. Results and Discussion

### 3.1. Evaluation of the Polymerization Process

In this study, a wine lees extract was used as a source of different phenolic compounds to acquire complex oligomers/polymers with new characteristics and sufficient biological activity. According to the literature, TvL catalyzes the oxidation of polyphenols with no substrate specificity [39]. Thus, TvL was used to oxidize the phenolics of the WLE towards the formation of new biopolymeric products.

As the first step, the activity and stability of TvL at the reaction conditions were determined to ensure that the enzyme is effective during the oxidation–polymerization process. The specific activity of TvL was determined at 0.096 ± 0.003 U mg^−1^. The stability results showed that TvL maintained 98.6 ± 1.1% of its initial activity for up to 24 h of incubation at 30 °C. In the next step, the free -OH group content of the reaction solution was estimated using the Folin–Ciocalteu assay. The assay can detect the free -OH of the phenolic compounds in a medium. After enzymatic oxidation, the phenolic compounds are converted to the corresponding quinones, leading to a free -OH content reduction. This reduction, referred to as dephenolization, was estimated based on the difference in the TPC before and after the enzymatic treatment, as previously described by Su et al. [18]. As it is observed in Table 2, there was no significant difference in the dephenolization yield between the different incubation times of the enzymatic reaction. The polymeric products recovery ranged from 22.0 to 25.7%, the latter corresponding to the highest amount of polymeric products formed after 24 h. This result indicates that most of the phenolics had been oxidized after 1 h of enzymatic treatment. Moreover, the mass of the produced polymeric products was slightly higher as the incubation time increased. This difference could be attributed to the second step of the procedure, the oxidative coupling. We can assume that as the spontaneous oxidative coupling of the formed quinones to oligomers or polymers is a chemical procedure, it is not affected by the presence of laccase [40].

The reaction progress was also monitored by UV-Vis spectroscopy. The UV-Vis spectra of the WLE and the supernatants after the treatment with TvL are presented in Figure 1. One main peak at 260–290 nm (with a maximum at 273 nm) and two broad bands at 350–400 nm and 420–650 nm can be identified in the extract’s spectra, as also previously reported for wine lees extracts [41,42]. The main peak at 260–290 nm is attributed to the presence of phenolic compounds, while the bands at 350–400 nm and 420–650 nm could be ascribed to flavonoids and anthocyanins, respectively [42]. As time increased, the color of the reaction changed from dark purple to dark brown, and polymeric products were observed to precipitate. These observations were mirrored by the gradual disappearance of the UV-Vis characteristic peaks after 10 min of enzymatic treatment. More specifically, the broad peaks presented at 350–400 and 420–650 nm of the extract were depleted after enzymatic treatment, indicating that flavonoids and anthocyanins were the first phenolic compounds to be oxidized by laccase. Similarly, the absence of these peaks was also confirmed in the UV-Vis spectra of the laccase-oxidized grape extracts, as reported by Li et al. [19]. No significant alterations could be observed after 1 h of incubation, indicating that the enzymatic oxidation reaction was completed in the first hour of treatment. A similar result was observed by Spyrou et al. where the peaks of the initial phenolic extract were decreased and broadened after enzymatic-assisted oxidation of the *Ulva* sp. extract [6]. The results of the UV-Vis analysis are in line with the results obtained from the Folin–Ciocalteu assay described above, where the dephenolization yields at 1, 2, 6, and 24 h were almost the same, hinting that the oxidation is almost completed at the first hour of enzymatic treatment.

The HPLC analysis allowed for an initial characterization of the phenolic profile of the WLE, as well as for the phenolic compounds that were involved in the polymerization process to be determined. Table 3 shows the main compounds that characterize the extract. Different known phenolic acids, phenolic alcohols, and a flavonoid were mostly present, which is in accordance with the literature, yet unknown polyphenols and a major fraction attributed to procyanidins/anthocyanins were also evident in the chromatogram (Appendix A) before the enzymatic treatment [34,43,44]. These fractions were not identified; however, these polyphenols also contribute to the formation of polymeric products. After the laccase-mediated polymerization, the peaks of all the known phenolic compounds decreased or disappeared, indicating an extensive reduction in the free phenolic compounds’ concentration (Table 3, Appendix A). The appearance of new peaks could be assigned to the formation of polymeric structures (dimers, trimers, tetramers, or polymers of phenolic compounds) during the 1 h of incubation. Latos-Brozio et al. have also observed a similar pattern for the polymerization of specific flavonoid compounds [15]. These results may explain the results obtained from the Folin–Ciocalteu assay and the UV-Vis spectroscopy, which indicate the almost complete oxidation of the main phenolic compounds after 1 h of treatment.

An NMR study also confirmed the oxidation of the phenolic compounds of the WLE. As shown in Figure 2, after the enzymatic oxidation reaction, a significant decrease, disappearance, or shift in the peaks in the range of 6–8.30 ppm was observed. Hence, the aromatic hydroxyl groups were involved in the polymerization process. Previous studies have also ascertained the participation of these groups in oxidative coupling [15,18,45,46]. As Latos-Brozio et al. also stated, minor differences were also observed in the range of 3–5 ppm that can be attributed to C−C bonds between flavonoid compounds which occurred during the polymerization process [15,45].

FTIR spectroscopy was also applied to evaluate the formation of the polymeric products concerning the reaction time (Figure 3). It can be observed that the spectra of the polymers differentiate from the spectrum of the extract. More specifically, the band of the extract at the region of 1710–1730 cm^−1^ corresponding to C=O stretching vibrations of the carboxyl groups, was diminished as the formation of the polymers took place. The band of the extract at 1625 cm^−1^, which is correlated to the aromatic C=C stretching vibrations, was shifted to higher wavenumbers (up to 1641 cm^−1^) after 24 h of enzymatic treatment, indicating alterations in the aromatic rings of the phenolic compounds due to the polymerization reaction. At higher time intervals, the appearance of a new band at 1530 cm^−1^ was observed and is ascribed to the presence of *o*-quinones that are formed after the enzymatic treatment [15]. As incubation time increased, the band at 1110 cm^−1^, ascribed to phenolic C-O vibrations, was eliminated, suggesting the appearance of the oxidative coupling reactions [45], while the appearance of a band at 1230 cm^−1^ is associated with the phenolic C-O stretching vibrations [47]. Lastly, the band at 1060 cm^−1^ which appeared in the 24 h polymer corresponds to the ketone groups (C–CO–C), indicative of the polymerization process [15,45]. The FTIR results reinforce the hypothesis that while the enzymatic oxidation of the phenolic compounds is completed at the first hour of incubation, the oxidative coupling (the chemical step of the reaction) that leads to the formation of the polymers continues for up to 24 h.

Finally, the synthesized polymeric products and extract were also submitted to GPC analysis (Figure 4a,b). Significant differences were observed when comparing the relevant chromatograms. Firstly, the polymers’ chromatograms were clearer, resembling typical polymer GPC curves in contrast to the extract. No peaks were monitored in the region of 14–16 mL in the polymers’ chromatograms. In contrast, an irregular morphology with broad, indistinct peaks was observed in the extract in the same region. In the open literature, there are various works where different natural substances (e.g., extracts from lignocellulosic materials [48], waste woody materials [49], agro-industrial residues [50], and phenolic substances [51]) have been submitted to GPC and similar curves with broad and/or multiple peaks have been obtained.

Focusing on the synthesized polymers’ chromatograms, unreacted extract residues were observed with the relevant peaks at 17–19 mL corresponding to MW values of 300 to 1900 g mol^−1^. At lower elution volumes (5–14 mL), three distinct, sharp peaks indicating three different higher MW populations were detected for the P24h corresponding to 44 × 10^3^, 20 × 10^4^, and 10^6^ g mol^−1^. On the contrary, the P1h, P2h, and P6h presented broad, unsplit peaks corresponding to 33 × 10^4^, 29 × 10^4^, and 45 × 10^4^ g mol^−1^, respectively. Thus, the increased reaction time (24 h) favored the polymerization and led to products of notably higher MW values.

Macroscopically, the polymeric products presented a uniform morphology compared to the extract, which seemed inhomogeneous, probably due to the presence of different compounds in its mass (an irregular morphology with broad indistinct peaks was observed in the extract GPC chromatogram). All the polymeric products (P1h, P2h, P6h, P24h) were received in the form of brown, free-flowing powders; P1h is indicatively presented in Figure 5a. The solid-state character and the free-flowing morphology indicate high polymerization degrees, in line with the GPC-derived results. On the contrary, the freeze-dried extract presented a sticky morphology with aggregated particles, attributed to the presence of low-molecular-weight components; the extract also presented a dark purple color (Figure 5b). The different colors of the polymeric products (brown) and the extract (dark purple) indicate their different chemical compositions. The purple color of the extract can be ascribed to anthocyanins existing in red grapes and wines as complexes either with themselves (self-association) or with other compounds, resulting in the formation of co-pigments. These are formed by processes involving stacked molecular aggregation, primarily held together by hydrophobic interaction; they can significantly increase color density (hyperchromic effect) and affect color tint, giving a more purple hue [52]. Regarding the polymer’s brown color, quinones, produced by polyphenol oxidases (i.e., laccases in our case), are very reactive species and electrophiles that undergo (non-enzymatic) polymerization to yield brown-colored polymeric pigments and by-products [53,54]. The color differences between the polymers and the extract were also observable when the samples were dissolved in DMF; the polymers’ solutions were yellowish, while the extract was brownish (Appendix A), with better solubility. The extract’s higher susceptibility to dissolution confirms the presence of lower molecular weight components compared to the polymers, as indicated via GPC.

### 3.2. Thermal Analysis of the Formed Polymers

Different behavior was observed during TGA between the extract and the synthesized products for reaction times of 1 and 24 h: the extract’s weight loss began already after 150 °C (*T*_d,5%_ = 164 °C), while the polymers presented slight weight loss at 30–100 °C, due to residual water or solvents, and then were thermally stable up to 180 °C with *T*_d,5%_ being increased with reaction time up to 186 and 192.5 °C for the P1h and P24h, respectively (Figure 6a, Appendix A). Given that *T*_d,5%_ is strongly affected by polymer MW, this trend aligns with the GPC results: a higher MW of the P24h compared to the P1h. On the other hand, the residues at 600 °C were found to be similar for all three samples, ranging between 29 and 35% (Appendix A). These high char residues are typical for phenolic compounds due to the aromatic rings in their structures; for instance, polycatechol and poly(gallic acid) degradation in TGA (30–800 °C, under N_2_) has been reported in the literature to result in char residues of ca. 57 and 25%, respectively [55]

Another noticeable difference between the extract and the polymeric products is their decomposition profile (Figure 6b). The extract presented a single-step decomposition (*T*_d_ = 194 °C) with two shoulders at ca. 300 and 350 °C. The herein-used extract originates from wine lees. In different works, winery lignocellulosic wastes (e.g., vine shoots and wine pomaces) have been submitted to TGA under N_2_. Among their substances, pectin and non-structural sugars have been shown to decompose at 240–270 °C, hemicellulose at ca. 300 °C, cellulose at ca. 340 °C, and lignin after 400 °C [56,57].

On the other hand, a two-step decomposition at higher temperatures (*T*_d1_ 218–238 °C, *T*_d2_ 271–308 °C) occurred for the P1h and P24h. This decomposition profile can be correlated to the different MW fractions detected in the GPC. Accordingly, in the literature, polyesters including poly(butylene succinate) (PBS) [58,59], poly(ethylene succinate) (PES) [60], and poly(ε-caprolactone) (PCL) [61] have been found to decompose in multiple steps when their MW is below a critical value, showing that some decomposition reactions are restricted to the lower molecular weight fractions probably due to increased end-group concentrations. Regarding the effect of polymerization time, for P24h, the second decomposition peak seems to dominate, as it is broader and more intense compared to the first peak, in contrast to P1h, where the opposite occurs with the first peak dominating (Figure 6b). This difference could indicate a higher MW population with increased thermal stability for the P24h compared to the P1h, in line with GPC results and the above-mentioned *T*_d,5%_ values.

Turning to the DSC results, the endotherm monitored at 30–40 °C in all samples (Figure 7, Appendix A) confirmed the presence of residual water and/or solvents, also evidenced in the polymers’ TGA. A remarkable difference between the polymers and the extract is an endotherm detected at 122–123 °C only for the polymers (Figure 7b), probably indicating melting since no weight loss was monitored at this temperature in TGA. This transition, absent from the extract thermogram (Figure 7a), further confirmed the successful formation of polymers. Two more broad endotherms were observed in the products’ thermograms, with the first at 220–300 °C and the second at 320–380 °C. These peaks correspond to the products’ two-step degradation, also monitored in TGA. Regarding the extract DSC, multiple indistinct endotherms were detected after 125 °C and up to 360 °C (Figure 7a), coinciding with its degradation observed in TGA (see Figure 6).

Overall, the herein-conducted polymerization can be considered successful based on the products’ defined thermal properties (increased thermal stability compared to the extract, thermal transition at 122 °C monitored only for the polymers) and the obtained GPC results (clearer chromatograms of the products, similar to typical polymer curves, high MW values). Regarding the examined reaction times (1, 2, 6, and 24 h), the P24h was superior, with a higher *T*_d,5%_, in agreement with its higher MW, and the second decomposition peak dominating in the TGA curve, in contrast to the P1h.

### 3.3. Antioxidant Activity of the Formed Polymers

The antioxidant activity of the obtained polymers was evaluated by ABTS and DPPH free radicals. The results were expressed as TE (Trolox being a standard synthetic antioxidant), as well as % antioxidant activity using a concentration of 100 μg mL^−1^ polymeric products (Appendix A). The antioxidant capacity of the polymeric products was up to ~2.4-fold and ~4-fold lower than the corresponding extract, according to DPPH and ABTS assays, respectively. This result could be attributed to the loss of functional -OH groups due to the oxidation and polymerization process. Similar results were observed for the enzymatic oxidation and polymerization process of phloridzin [62]. The authors reported that the IC_50_ value of the polymeric phloridzin was ~2.5-fold lower than the corresponding monomer. Furthermore, according to Latos-Brozio et al., the catechin polymers obtained from laccase or peroxidase treatment led to lower antioxidant capacity than their respective monomers due to the limitation of the reaction of active groups, such as –OH, with the free radicals [15]. According to the authors, the poly-catechin obtained after laccase treatment exhibited ~4-fold and 26-fold reductions in the ABTS and DPPH radicals, respectively, at a tested concentration of 100 μg mL^−1^, compared to catechin. As shown in Table 4, the P1h demonstrated the highest antioxidant activity. The other polymer products exhibited lower values, indicating lower antioxidant activity with no observed significant difference among them. These findings are similar to those reported for the antioxidant activity of the products derived from the laccase-catalyzed oxidation of propolis and poplar bud extracts [25]. The authors reported that the IC_50_ values of the oxidized products were 35, 99, and 294 μg/mL after 1, 7, and 15 h of laccase-catalyzed oxidation, respectively, indicating that the antioxidant activity decreases as the reaction time increases [25]. The highest antioxidant capacity observed for P1h could be attributed to the smaller polymer consistency. As evaluated from the GPC analysis, the fractions of P1h have the lowest molecular weight among the four products. It is expected that small compounds can react more efficiently with the DPPH and ABTS free radicals than bulky compounds due to steric hindrance phenomena arising from the latter. Accordingly, the polymers with higher molecular weights may not be easily accessible to the free radicals [63,64,65,66]. For instance, Xiao et al. reported that the dimerization of flavonols leads to lower antioxidant activity due to higher steric hindrance of the dimer structure [65].

### 3.4. Properties of the Polymer-Enriched CS Films

#### 3.4.1. Antioxidant Activity of the CS-Polymer Films

The antioxidant activity of the phenolic polymer-enriched CS films is presented in Figure 8. It must be mentioned that when the same proportion of WLE was loaded in the CS films (20 mg), the antioxidant capacity of the CS-extract film using the DPPH assay was ~52%. This antioxidant activity is similar to or lower than that of the same loaded CS-polymers films. As seen, the addition of the phenolic polymers improved the antioxidant potential of the films, exhibiting antioxidant activities of almost 80 and 70% for ABTS and DPPH, respectively. Moreover, the films containing higher amounts of the polymers presented higher antioxidant activity, demonstrating that the result is dose-dependent. The dose-dependence effect is very common in packaging films incorporating natural extracts rich in phenolic compounds. For example, the incorporation of curcumin, mango leaf, or vinasse extracts in natural polysaccharides packaging films increases the antioxidant activity of the films proportionally to the amount of added extract [30,67,68]. Moreover, CS films enriched with grape pomace or grape seed extracts exhibited 16 and 34% antioxidant activity, respectively, using DPPH. Similar results were obtained using the ABTS free radical, where the extracts exhibited 12 and 25% antioxidant activity, respectively [27]. These results indicate that our films are promising for the development of active food packaging alternatives. Finally, the DPPH results showed that the antioxidant activity of the films loaded with P1h was higher than the antioxidant activity of the films with the other polymers, indicating that P1h was the most effective as an improving agent for food packaging applications. This result is in line with the results from Table 4, where P1h exhibited the highest antioxidant activity.

#### 3.4.2. Moisture Content, Water Swelling, and Water Solubility of the Films

Chitosan is well known as a biopolymer for food packaging applications due to its safety, high biodegradability, and mechanical and water-sensitivity properties [26,38,69]. Moisture content, water swelling, and water solubility are important indicators of a film’s quality [38,69]. For the estimation of the MC, WSw, and WS values, the films enriched with P1h were chosen as they presented the highest antioxidant activity toward DPPH. The results are summarized in Figure 9. As observed, the addition of P1h does not significantly affect the water sensitivity of the films compared to the blank chitosan films. Similarly, the MC of the films remained unaltered after the addition of the polymer. On the contrary, the WS and WSw of the CS-P1h films were slightly increased as the amount of the polymer increased, except for the lowest loading concentration. However, these small changes are not considered significant in affecting the film’s properties. These observations for the water sensitivity tests indicate that the CS films are easily biodegradable even after the addition of the polymer [70,71]. The swelling degree of the films is correlated with their biodegradability; the permeation of water molecules into the films leads to film swelling, facilitating their enzymatic degradation by the soil microorganisms [71,72].

## 4. Conclusions

In this work, wine lees were exploited due to their high phenolic content to attain products with interesting properties. TvL was used as a biocatalytic tool for the oxidation of a wine lees phenolic extract to produce new phenolic polymers with tunable properties that could potentially replace their phenolic precursors in many applications, such as in food packaging films. In this context, the laccase-catalyzed oxidation of the WLE was investigated in various time intervals and different polymeric products were produced. The obtained results support that the laccase-catalyzed oxidation reaction is completed after 1 h of enzymatic treatment while the polymerization process continues for up to 24 h, as confirmed by different spectroscopic and analytical techniques. The synthesized polymeric products proved to be effective as additives in food packaging films, reinforcing their initial antioxidant activity. Summarizing, the results of this work reveal the potential use of laccases as biocatalysts to produce value-added polymers from wine lees as new bioactive materials for alternative packaging applications. This bioprocess could be applied to other by-products, as well, paving the way for sustainable solutions and promoting the principles of circular economy. However, there is still a great need for further investigation of the process and characterization of the isolated polymeric products to deepen our understanding of the reaction mechanisms and their impact on the properties of the final products.

## Data Availability

The data are contained within the article.

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
