# Peer review of "Laccase-Mediated Oxidation of Phenolic Compounds from Wine Lees Extract towards the Synthesis of Polymers with Potential Applications in Food Packaging"

_biomolecules, 2024, doi:10.3390/biom14030323_

Round 1

Reviewer 1 Report

Comments and Suggestions for Authors

Dear authors,

I would like to acknowledge your work presented in the manuscript in a clear, sound and readable manner. I have rated the manuscript with high marks. However, my recommendation is major review due to the one point I think is incomplete.

My main recommendation and doubt is related to antioxidant activity. I think that antioxidant activity of the extract is missing. If I have understood properly, we are trying to replace the extract with the polymer, but if we do not know what is the antioxidant activity of the extract and chitosan film containing extract, we are missing the perspective.

Also, in the section antioxidant activity of the polymers,  a comparison with known antioxidants that have a proven effect in application is lacking. This is important to get an insight if the obtained polymers are in the teoretical or practical scale of activity. Please provide comparation to the activity of other natural active extracts and polimeric forms, as well as active chitosan films. Also L520-521: "sufficient activity" sufficient for what and according to what criteria?

Why did you choose to present the two antioxidant activities in different units? In this way, extent of activity reduction after incorporation in chitosan film is not visible.

In addition, my impression is  that Table 4 should be a part of material and methods section

 Also, lines 463-470 are more appropriate for introductory section.

L502-504: Are you sure that water sensitivity directly relates to the biodegradability of the films?  

Best regards

Author Response

Dear authors,

I would like to acknowledge your work presented in the manuscript in a clear, sound and readable manner. I have rated the manuscript with high marks. However, my recommendation is major review due to the one point I think is incomplete.

  1. My main recommendation and doubt is related to antioxidant activity. I think that antioxidant activity of the extract is missing. If I have understood properly, we are trying to replace the extract with the polymer, but if we do not know what is the antioxidant activity of the extract and chitosan film-containing extract, we are missing the perspective.

We thank the reviewer for the valuable comment. Firstly, we would like to clarify that our main goal was the enzymatic production of stable polymeric products with antioxidant properties for potential use in food packaging films. These polymeric products may replace the extract during the synthesis of the packaging film by improving the stability of the films, as monomeric phenolic antioxidants could be released from the polymer matrix during contact with water. Moreover, we would like to note that we have already examined the antioxidant activity of the extract, which was up to 2.4-fold and 4-fold higher than that of the polymeric products according to DPPH and ABTS assays, respectively. The manuscript was revised to provide the antioxidant activity of the extract and CS-extract film (Please see Lines 520-523, 553-558).

  1. Also, in the section antioxidant activity of the polymers, a comparison with known antioxidants that have a proven effect in application is lacking. This is important to get an insight if the obtained polymers are in the teoretical or practical scale of activity. Please provide comparation to the activity of other natural active extracts and polimeric forms, as well as active chitosan films. Also L520-521: "sufficient activity" sufficient for what and according to what criteria?

Thank you for the comment. We would like to clarify that the antioxidant activity of the formed polymeric products is expressed in terms of Trolox equivalents to obtain a direct comparison with a common synthetic antioxidant (Please see lines 518-519). Moreover, we tried to compare our results with those in the existing literature, to meet the reviewer ‘s comment (Lines 523-531, 534-539, 564-567).

  1. Why did you choose to present the two antioxidant activities in different units? In this way, extent of activity reduction after incorporation in chitosan film is not visible.

As mentioned in our previous response, the antioxidant activity of the polymers is expressed in terms of TE to obtain a direct comparison with a common synthetic antioxidant. However, this kind of activity expression cannot be applied to the films, as they are not soluble or dispersed in the reaction media. Thus, we presented the antioxidant activity of the films as % antioxidant activity, being the most common expression in the relevant literature (e.g. https://doi.org/10.1016/j.carbpol.2014.07.032, https://doi.org/10.3390/coatings10100936, https://doi.org/10.1016/j.foodhyd.2016.04.032).

However, the antioxidant activity of the formed polymeric products was also presented as % antioxidant activity using a concentration of 20 μg/mL to meet the reviewer’s suggestion. The results can be found in Table S3 in the revised Supplementary Materials as indicated in the revised manuscript. (Lines 519-520).

  1. In addition, my impression is  that Table 4 should be a part of material and methods section

We agree with the reviewer. Table 4 is now re-named as Table 1 and is moved to the materials and methods section.

  1. Also, lines 463-470 are more appropriate for introductory section.

Thank you for the suggestion. We have transferred the paragraph in the introduction section (Lines 88-92).

  1. L502-504: Are you sure that water sensitivity directly relates to the biodegradability of the films?  

We would like to comment that according to the existing literature, water sensitivity is correlated to the biodegradability of the films. For instance, De Carli et al. reported that due to water swelling, the degradation of the films is promoted (https://doi.org/10.1016/j.ijbiomac.2022.05.155). Moreover, the water solubility is an indicator of the biodegradability. The manuscript was revised to contain a more precise explanation (Lines 588-590).

Reviewer 2 Report

Comments and Suggestions for Authors

Athanasiou and coworkers described a laccase-mediated oxidation of phenolic compounds to synthesise polymers with potential applications in food packaging. However, major revision are required to support the enzymatic approach they decided to follow.

Indeed, the authors often draw the conclusion that the enzymatic oxidation of the phenolic compounds is completed at the first hour of incubation and then other non-enzymatic couplings occur. However, no investigation about enzyme activity/stability was carried put during the reaction time. Residual activity of the enzyme can be instrumental to understand its role in the polymerization of phenols.

Some minor points:

-       -  Why did the authors enrich the chitosan film with all the synthesized polymers, while they already observed that P1h had displayed the highest antioxidant activity?

-        - There are some typos in the manuscript (fungal genus not abbreviated and text in red throughout the manuscript)

Author Response

Athanasiou and coworkers described a laccase-mediated oxidation of phenolic compounds to synthesise polymers with potential applications in food packaging. However, major revision are required to support the enzymatic approach they decided to follow.

Indeed, the authors often draw the conclusion that the enzymatic oxidation of the phenolic compounds is completed at the first hour of incubation and then other non-enzymatic couplings occur. However, no investigation about enzyme activity/stability was carried put during the reaction time. Residual activity of the enzyme can be instrumental to understand its role in the polymerization of phenols.

We thank the reviewer for the valuable contribution. We agree that enzyme stability is crucial for the described process, thus we have conducted stability experiments. The results showed that TvL was stable for 24 h with the residual activity being 98.6 ± 1.1 %. The manuscript was revised according to preset this information (Please see lines 141-154, 311-315).

Some minor points:

-  Why did the authors enrich the chitosan film with all the synthesized polymers, while they already observed that P1h had displayed the highest antioxidant activity?

We would like to clarify that all the synthesized polymers were used as additives in chitosan films to confirm that the P1h-enriched films would exhibit higher antioxidant activity than the films enriched with the other polymeric products.

- There are some typos in the manuscript (fungal genus not abbreviated and text in red throughout the manuscript)

We have revised the manuscript to correct any typos.

Reviewer 3 Report

Comments and Suggestions for Authors

This manuscript has many necessary points should be improved, but it may be reconsidered after major revisions.

1. The introduction may lack an introduction to the relevant and latest application developments to provide a good overview of the development in this field.

2. L58: The information presented in Scheme 1 is insufficient to reveal polymerization mechanism of phenolic compounds. In fact, it only briefly lists some possible monomers but does not reveal how these monomers polymerize.

3. L81~ 95: Overly detailed introductions to the methods here are unnecessary. Place research objectives and characteristics instead.

4. Sufficient details of the used materials, especially Laccase, Trametes Versicolor, Wine Lees, etc, should be provided to allow for the replicability of the work,

5. L126: It lacks detailed information on wine lees in the manuscript. Obviously, their extracts are an extremely complex mixture of ingredients. It is highly questionable whether it will undergo some predictable polymerization reactions and form polymers under a relatively mild condition.

6. The authors did not provide the macroscopic and microscopic morphology of the final polymer products, which can provide a lot of information, such as whether the final product is a collection of incompatible and complex components.

7. L209: More detailed information on chitosan should be provided, such as supplier, deacetylation degree, viscosity, polymerization degree, etc.

8. L211~212: List these loading concentrations here.

9. Does Amount of produced polymer in Table 1 refer to the quality of the final product? This is obviously appropriate, as the final products are a complex collection of unknown components. Obviously, it cannot be simply regarded as a polymer. On this basis, the calculation of Polymerization yield is also meaningless.

10. L294~296: The expression here is quite vague and seems incorrect, as the peaks at 350~400 nm and 420~650 nm seems only appear at time 0.

11. It is inconvenient to combine Table 2 and Figure S1. A better method is to label the corresponding components and their reductions on the corresponding peaks on the HPLC curve.

12. Why thermodynamic analysis lacks data of P2h and P6h?

13. L463~473: More suitable for Introduction section

14. Table 4 is more suitable for replacing to Materials and Methods section.

15. Results and discussions section only focus on listing experimental data, lacking sufficient analysis of the in-depth mechanisms behind the experimental results, as well as sufficient comparison with similar works.

16. Conclusions should be rephrased, without the need for detailed introduction of experimental methods and results, but more conclusions based on experimental data are necessary.

Comments on the Quality of English Language

Minor editing of English language required.

Author Response

This manuscript has many necessary points should be improved, but it may be reconsidered after major revisions.

  1. The introduction may lack an introduction to the relevant and latest application developments to provide a good overview of the development in this field.

We thank the reviewer for the comment. The introduction was rephrased and enriched with more information about relevant works, as well as more information about active packaging films to meet the reviewer's suggestion (Lines 72-83, 94-98).

  1. L58: The information presented in Scheme 1 is insufficient to reveal “polymerization mechanism of phenolic compounds”. In fact, it only briefly lists some possible monomers but does not reveal how these monomers polymerize.

Scheme 1 was revised to include all the possible substrate combinations for the polymerization step. Moreover, the manuscript was revised to present Scheme 1 in a better way (Lines 50-55).

  1. L81~ 95: Overly detailed introductions to the methods here are unnecessary. Place research objectives and characteristics instead.

The manuscript was revised according to the reviewer’s suggestions (Lines 104-116).

  1. Sufficient details of the used materials, especially Laccase, Trametes Versicolor, Wine Lees, etc, should be provided to allow for the replicability of the work

We would like to clarify that all the details of the used chemicals and reagents are described in section 2.1 Chemicals and Reagents (Line 119 about Laccase from Trametes versicolor), while the details for the wine lees are described in section 2.2 Wine lees extraction (Lines 113-115).

  1. L126: It lacks detailed information on wine lees in the manuscript. Obviously, their extracts are an extremely complex mixture of ingredients. It is highly questionable whether it will undergo some predictable polymerization reactions and form polymers under a relatively mild condition.

Thank you for the comment. We agree with the reviewer that our extract is a complex mixture of ingredients. An estimation of the phenolic profile of the wine lees extract was provided by HPLC analysis (Please see Figure S1). We also agree that the prediction of the polymerization reaction is not possible in such a complex sample. However, as hinted by HPLC analysis, after the enzymatic treatment most of the phenolic compounds have been decreased (Table 3, Figure S2), indicating that these compounds take part in the formation of the polymeric products. Moreover, NMR analysis (Lines 379-386, Figure 2) revealed the reduction of the aromatic hydroxyl groups in the supernatant after the polymerization process, indicating that phenolic compounds were involved in the polymerization process. The NMR spectra of the polymeric products also show that a part of the aromatic hydroxyl groups was still presented, confirming that these products partly consisted of aromatic compounds. Furthermore, the polymeric forms were characterized by GPC (Lines 425-432, Figure 4), confirming that the obtained precipitates consisted of compounds with high molecular weights.

  1. The authors did not provide the macroscopic and microscopic morphology of the final polymer products, which can provide a lot of information, such as whether the final product is a collection of incompatible and complex components.

We want to thank the reviewer for the valuable comment. We agree that the macroscopic and microscopic characterization of the extract would increase the quality of our work. Macroscopic characterization is now provided in the revised manuscript (Lines 433-456, Figure 5, Figure S3). Due to the limited revision time and the limited access to these techniques, it was not possible to carry out the microscopic characterizations.

  1. L209: More detailed information on chitosan should be provided, such as supplier, deacetylation degree, viscosity, polymerization degree, etc.

Detailed information on chitosan is provided in the revised manuscript (Line 120).

  1. L211~212: List these loading concentrations here.

The concentrations are now listed in Table 1 of the revised manuscript.

  1. Does “Amount of produced polymer” in Table 1 refer to the quality of the final product? This is obviously appropriate, as the final products are a complex collection of unknown components. Obviously, it cannot be simply regarded as a “polymer”. On this basis, the calculation of “Polymerization yield” is also meaningless.

Thank you for the comment. We agree that the final product of each enzymatic treatment (P1h, P2h, P6h, and P24h) is a complex mixture of unknown components. This is also confirmed by GPC results. To avoid misunderstandings, we decided to refer to these components as “polymeric products”, and we used this term throughout the manuscript.

We would also like to clarify that the amount of the produced polymeric products refers to the quantity and not to the quality of the final product, while the polymerization yield is a recovery indicator for the quantity of the polymeric products concerning the original extract, as now indicated in the revised manuscript (Table 3, Line 322).

  1. L294~296: The expression here is quite vague and seems incorrect, as the peaks at 350~400 nm and 420~650 nm seems only appear at time 0.

Thank you for the comment. The peaks at 350-400 nm and 420-650 nm correspond to the presence of flavonoids and anthocyanins, respectively, as already mentioned in the manuscript. These two bands disappear after 10 min of enzymatic treatment, indicating that these two phenolic groups are the first to be oxidized by laccase. Similar observations were reported in another published work (https://doi.org/10.1021/acs.jafc.3c04798). The manuscript was revised to provide a more detailed explanation of the depleted peaks (Lines 344-349).

  1. It is inconvenient to combine Table 2 and Figure S1. A better method is to label the corresponding components and their reductions on the corresponding peaks on the HPLC curve.

We want to thank the reviewer for the valuable comment. We have included labels in Figure S1 in the revised supplementary materials and provided an additional Figure (Figure S2) which illustrates the effect of the enzymatic treatment on the reduction of the peaks of individual phenolic compounds. The data of Table 2 (re-named as Table 3 in the revised manuscript), Figure S1, and Figure S2 are complementary and exhibit an overview of the first reaction step.

  1. Why thermodynamic analysis lacks data of P2h and P6h?

Thank you for the comment. We decided to conduct a thermodynamical analysis for P1h, as this product displayed higher antioxidant activity than the other polymeric products. In the next step, we chose to analyze the P24h too because we wanted to compare the formed products derived from the lowest and highest incubation time of enzymatic treatment.

  1. L463~473: More suitable for Introduction section

We would like to thank the reviewer for the suggestion. The paragraph was moved to the introduction section (Lines 88-92).

  1. Table 4 is more suitable for replacing to Materials and Methods section.

We agree with the reviewer. Table 4 is now re-named as Table 1 and is moved to the materials and methods section.

  1. Results and discussions section only focus on listing experimental data, lacking sufficient analysis of the in-depth mechanisms behind the experimental results, as well as sufficient comparison with similar works.

 The manuscript was revised focusing on the comparison of our work with other works. Moreover, an additional discussion of the enzymatic oxidation and polymerization process described in the present work is included in the revised manuscript (e.g. Lines 344-349, 523-531, 534-539, 564-567).

  1. Conclusions should be rephrased, without the need for detailed introduction of experimental methods and results, but more conclusions based on experimental data are necessary.

Conclusions were revised according to the reviewer’s suggestions (Lines 594-611).

Round 2

Reviewer 1 Report

Comments and Suggestions for Authors

Dear authors,

thank you for considering all the suggestions and for providing appropriate and satisfying replies.

I have still one unclear detail. That is, how do you suppose that polymer with activity up to 40% by itself reached in chitosan film activity up to 70-80%?

Author Response

Dear authors,

thank you for considering all the suggestions and for providing appropriate and satisfying replies.

I have still one unclear detail. That is, how do you suppose that polymer with activity up to 40% by itself reached in chitosan film activity up to 70-80%?

We thank the reviewer for the valuable feedback. The antioxidant activity of the polymer is dose-dependent; it increases with the increase of the polymer concentration. High polymer concentrations (100 μg/mL) led up to 93% antioxidant activity. The manuscript was revised according to the reviewer's comment (Lines 523-524) and the concentration of 20 μg/mL was replaced by the highest tested concentration of polymers (100 μg/mL) in Table S3 of the supplementary materials. Moreover, we would like to clarify that the antioxidant activity of the polymers and the antioxidant activity of the polymer-enriched films cannot be directly correlated. The interactions between the incorporated polymer and the free radical are restricted, due to its entrapment in the chitosan matrix, thus leading to lower antioxidant activity. Furthermore, the films are not soluble or dispersed in the reaction media to reach the same antioxidant activity as the free polymers.

Reviewer 2 Report

Comments and Suggestions for Authors

Although the authors assessed enzyme stability, results may not be relevant in the reaction condition, since they performed stability experiments in buffer and 30°C without adding phenol extracts to the mixture. Enzyme activity has to be assessed by measuring ABTS oxidation in the same conditions of polymer synthesis, since the substrate itself or some products may affect enzyme activity.

Author Response

Although the authors assessed enzyme stability, results may not be relevant in the reaction condition, since they performed stability experiments in buffer and 30°C without adding phenol extracts to the mixture. Enzyme activity has to be assessed by measuring ABTS oxidation in the same conditions of polymer synthesis, since the substrate itself or some products may affect enzyme activity.

We thank the reviewer for the valuable comment. We would like to clarify that we had already tried to measure the laccase activity in the presence of the phenolic extract, by measuring ABTS oxidation. Unfortunately, the enzyme activity cannot be assessed by this protocol. The phenolic compounds of the extract are natural substrates of laccase, while ABTS acts both as a substrate and mediator for the oxidation of the phenolic compounds, making the determination of the reaction rate not possible. Based on these results and observations, we decided to present the activity and stability results in the absence of the extract.

Reviewer 3 Report

Comments and Suggestions for Authors

The manuscript has made some changes, but at least two responses are very unsatisfactory. Firstly, the changes in Scheme 1 are meaningless as it still does not thoroughly reveal the "polymerization mechanism of phenolic compounds". Secondly, the manuscript still lacks methods for obtaining wine lees. A basic requirement for a published research work is the reproducibility of the method. The lack of production methods for a key material obviously cannot meet the requirements of publication. HPLC analysis is only a characterization of it and cannot be used to infer the production method in reverse.

Comments on the Quality of English Language

 Minor editing of English language required.

Author Response

The manuscript has made some changes, but at least two responses are very unsatisfactory.

1) Firstly, the changes in Scheme 1 are meaningless as it still does not thoroughly reveal the "polymerization mechanism of phenolic compounds".

We believe that the modified Scheme 1 in the revised manuscript (Lines 54-55, 63-65) is indicative of laccase-promoted polymerization reactions, and it follows other works in the existing literature (https://doi.org/10.3390/polym10101141, https://doi.org/10.1016/j.eti.2016.04.001, https://doi.org/10.1007/s12010-013-0463-0, https://doi.org/10.3390/life13020291).

2) Secondly, the manuscript still lacks methods for obtaining wine lees. A basic requirement for a published research work is the reproducibility of the method. The lack of production methods for a key material obviously cannot meet the requirements of publication. HPLC analysis is only a characterization of it and cannot be used to infer the production method in reverse.

All the obtained information about the production of wine lees is provided in the revised manuscript (Lines 133-140).